# The Joint Evolution of Herbivory Defense and Mating System in Plants: A Simulation Approach

**DOI:** 10.3390/plants12030555

**Published:** 2023-01-26

**Authors:** Edson Sandoval-Castellanos, Juan Núñez-Farfán

**Affiliations:** 1Laboratorio de Genética Ecológica y Evolución, Departamento de Ecología Evolutiva, Instituto de Ecología, National Autonomous University of Mexico, Mexico City 04510, Mexico; 2Population Genomics Group, Department of Veterinary Sciences, Ludwig Maximilian University of Munich, 82152 Munich, Germany

**Keywords:** agent-based model simulations, coevolution, gene-for-gene model, herbivory, inbreeding, mating system, resistance, selfing, tolerance

## Abstract

Agricultural losses brought about by insect herbivores can be reduced by understanding the strategies that plants use against insect herbivores. The two main strategies that plants use against herbivory are resistance and tolerance. They are, however, predicted to be mutually exclusive, yet numerous populations have them both (hence a mixed defense strategy). This has been explained, among other alternatives, by the non-linear behavior of the costs and benefits of resistance and tolerance and their interaction with plants’ mating system. Here, we studied how non-linearity and mating system affect the evolutionary stability of mixed defense strategies by means of agent-based model simulations. The simulations work on a novel model that was built upon previous ones. It incorporates resistance and tolerance costs and benefits, inbreeding depression, and a continuously scalable non-linearity. The factors that promoted the evolutionary stability of mixed defense strategies include a multiplicative allocation of costs and benefits of resistance and tolerance, a concave non-linearity, non-heritable selfing, and high tolerance costs. We also found new mechanisms, enabled by the mating system, that are worth considering for empirical studies. One was a double trade-off between resistance and tolerance, predicted as a consequence of costs duplication and the inducibility of tolerance, and the other was named the *resistance-cost-of-selfing*, a term coined by us, and was derived from the duplication of costs that homozygous individuals conveyed when a single resistance allele provided full protection.

## 1. Introduction

Between 20 to 40% of global crop production is lost to pests every year. Those losses cost >$200 billion annually, of which >$70 billion are due to invasive insects alone [1]. A better understanding of the strategies that plants use against insect herbivores could help to reduce these losses without increasing pesticide use [2]. The two main strategies that plants use against herbivores are resistance, the plant’s capacity to prevent herbivory, and tolerance, the capacity to reduce fitness losses brought about by herbivory [3]. Resistance traits include spines, wax, or trichomes that prevent/reduce consumption by physical means, and secondary specialized metabolites such as alkaloids, terpenes, glucosinolates, etc., that are toxic, repellent, or antidigestive to herbivores [4]. On the other hand, tolerance is thought to consist of traits that are associated with metabolism, resource reallocation, and growth. Resistance and tolerance traits have both been demonstrated to be highly heritable [3].

Both strategies have also been shown to exhibit fitness costs, making their simultaneous presence in a population (hence termed a mixed defense strategy) theoretically disadvantageous by duplicating defense costs [3,5]. In spite of that, mixed strategies are frequent in natural populations [3], which was initially explained by considering those populations in a transition towards a single strategy [5,6]. However, studies have accumulated evidence of factors that allow mixed strategies to be evolutionarily stable. Those factors include several types of selection [7,8,9], the local adaptation of herbivores [10,11], a non-linear accumulation of costs and benefits of the defense strategies [12], and plants’ mating system [3,13,14,15,16]. Here, we consider the effects of the non-linearity while focusing on the effect of the plant’s mating system, a topic that has gained prominence in recent years (e.g., [15,16,17,18,19]).

The mating system is the ratio of selfing and outcrossing in plant populations [20] and affects fitness through inbreeding depression [21,22]. Genes promoting selfing would bring about inbreeding depression costs, but also the “automatic selection advantage” that results from their two-fold representation in the offspring [21,23]. The final effect that selfing has on fitness usually involves the interactions between other factors such as resistance and tolerance and the mating system.

The mating system can affect resistance and tolerance by reducing the genetic variation needed for effective responses to herbivory [3]. Conversely, herbivores can alter a plant’s mating system by differentially affecting self- and cross-progenies in traits such as survival, vigor, floral display size, attractiveness to pollinators, pollen production, or flower availability [13,14,24,25,26,27,28,29]. Empirical studies have found a negative correlation between inbreeding and both resistance and tolerance [29,30,31,32,33,34], but the specific mechanisms behind those correlations are mostly unknown [3]. However, it is safe to state that plant defense strategies have the potential to be affected by plants’ mating systems (see reviews [3,15,16]).

On the other hand, theoretical studies have found that the shape of the functions modeling the fitness dependency on defense costs and benefits could be mostly non-linear [3,8,35,36]. Non-linearity could reduce the effective redundancy of the fitness benefits of resistance and tolerance, expanding the chances for mixed defense strategies [8,37].

Here we studied in silico the effect of the mating system on the evolutionary stability of the mixed defense strategies of plants. For this, we created a mathematical model that takes into account scalable non-linearity and implemented it in agent-based model (ABM) computer simulations to discover mechanisms of interaction between defense strategies and the plant’s mating system. This avoids the excessive complexity that may affect some empirical studies, as well as the oversimplification of purely theoretical approaches. With this strategy, we discovered several factors that favored the presence of mixed defense strategies and discussed their presence in natural populations.

## 2. Results

### 2.1. Equations’ Analysis and Predictions

#### 2.1.1. Costs and Benefits of Resistance and Tolerance

Equations (1) and (2), see Methods, predicted two alternative scenarios that maximized fitness (*W*_M_) as a function of the costs of resistance and tolerance (and tolerance benefit). The ratio of the maximum fitness obtained with the scenarios, *f_r_*, determined which of the theoretical scenarios would be selectively favored. For *f_r_* > 1.0, a non-resistance scenario ($w_{M}^{NRS}$) would be favored, while for *f_r_* < 1.0, a scenario with resistance ($w_{M}^{RS})\;$ would be (see a summary of predictions in Table 1). None of the scenarios predicted a mixed strategy, even when resistance and tolerance were not costly. This was due to a *double trade-off* between resistance and tolerance. The first trade-off came from the duplication of costs, and the second from the inducibility of tolerance that made its benefits dependent on herbivory damage, which, in turn, required the absence of resistance.

Figure 1 and Appendix A show the fitness surface predicted by Equations (1) and (2) as a function of both the costs and investments of tolerance and resistance. Here, the tolerance benefit was set to *b_t_* = 1 − *c_t_* (*c_t_* = tolerance cost) in order to assess the effect of the tolerance benefit and cost at once. A negative correlation between fitness and both costs and investments of resistance and tolerance was observed in the absence of herbivory. When herbivory was present, fitness increased with lower resistance because the resistance benefit is absent in the equations. However, fitness was sensitive to the balance of costs and benefits of tolerance.

The two scenarios, resistance and non-resistance, led to different expectations for the plant’s mating system. In the resistance scenario, resistance costs promote the minimization of the number of resistance alleles carried out by plants, as one single allele suffices to confer protection under the gene-for-gene model. In selfers, however, the increased homozygosis causes plants to carry more than one resistance allele, giving them additional resistance costs, which we call “*the resistance cost of selfing*”. Tolerance, in contrast, is not predicted to interact directly with the mating system. Despite these predictions, real populations can have a continuous supply of non-optimal genotypes by means of segregation and random mating, allowing the presence of mixed strategies. This, however, is better assessed by simulations.

#### 2.1.2. Inbreeding Depression

Some obvious predictions, e.g., the intensity of inbreeding depression being correlated with the selection coefficient of harmful alleles, will not be further discussed. A more useful task is the assessment of equilibrium points at which the systems would no longer change in terms of the number of harmful alleles and individual heterozygosity. The comparison between the predicted values and those observed in the simulations would help to identify unpredicted mechanisms that affect the mating system.

Under the dominant model, inbreeding depression would stabilize when the number of harmful alleles created by mutation equals their purging: 2*h_om_* = *U* (*h_om_* = individual homozygosis of harmful mutations; *U* = individual genomic mutation rate). Appendix A details the estimation of the cumulative number of lethal recessives in outcrossing and selfing plants. For the default values used in the simulations, the equilibria corresponded to 13.65 harmful alleles per individual for outcrossing plants and 1.0 harmful alleles for selfing plants.

For overdominance, inbreeding depression in selfing plants would stabilize when the selfing surge of homozygosis equaled the purging of lethal alleles (which were always recessive). For default values, the equilibrium occurred at *h_i_* = 0.01. Note that individual heterozygosis can be zero in the additive model but not in the multiplicative model where it cancels the entire fitness.

#### 2.1.3. Linearity

The linearity parameter *l* continuously shapes the fitness function making it convex or concave (see Table 2). The fitness functions represented by Equations (1) and (2) would display a concave fitness surface if they were considered solely as functions of the costs of resistance and tolerance and *l* > 1.0. Because a convex function is curved downwards, our equations would yield higher fitness than a linear (flat) function for most combinations of costs. In consequence, resistance and tolerance would be less costly, increasing the opportunity for mixed strategies. For *l* < 1.0, the fitness surface would be convex, and resistance and tolerance would be costlier, diminishing the opportunity for mixed strategies.

The effect of linearity becomes more complicated when Equations (1) and (2) are seen as functions of the herbivory damage and tolerance benefits (HD and TB, respectively). The relevant terms (*c_h_*, *d*_(*r*, *a*)_, *b_t_*, *t*) can have different signs, and a change in sign can turn convexity into concavity and vice versa, resulting in a composite behavior of the fitness surface. The fitness surface can even have a simultaneously convex and concave shape in different directions (see Figure 2). When this complex behavior occurs, the convexity of one term of the equation can cancel out the concavity of another one, yielding a flat surface. In spite of such complex behavior, it is more likely to have a concave surface when *l* > 1.0 for intermediate values of resistance cost, tolerance cost, tolerance benefit, herbivory damage, and herbivory cost; and a convex surface when *l* < 1.0 for those values. Recall that overall concavity promotes mixed strategies, and convexity diminished them. 

### 2.2. Simulation Results

Selfing and the type of model showed the deepest consequences in the evolution of the simulated systems. Appendix A display charts of the alleles’ *persistence* (the allele frequency averaged over the 15,000 simulated generations and 3 replicates) as a function of costs of resistance and tolerance, and tolerance benefit (*c_r_*, *c_t_*, and *b_t_*), as well as inbreeding depression intensity—as measured by parameters *k* and *s*—and the linearity parameter (*l*). Figures also show charts for each model type (multiplicative and additive) and type of selfing (heritable and non-heritable).

#### 2.2.1. Multiplicative vs. Additive Model

We obtained a fitness surface that was convex under the multiplicative model and flat under the additive model, as a function of costs (Appendix A). Because of this, the cost burden was less severe within the multiplicative model than within the additive model resulting in the increased persistence of resistance and tolerance and a chance of mixed strategies under the multiplicative model (top charts Appendix A). The boost to resistance that was promoted by the multiplicative model indirectly reduced heritable selfing due to the *resistance cost of selfing* (see “Equations’ analysis and predictions” above), which, in turn, promoted mixed mating systems.

While multiplicativity boosted the persistence of resistance, tolerance, and mixed mating systems, it was also able to cancel them completely, i.e. nullify the fitness if only one of the five terms of Equation (1) became zero. For instance, a lack of individual heterozygosis would be lethal under the multiplicative model but not for the additive model. This resulted in inbreeding depression becoming more severe under multiplicativity, decreasing heritable selfing and quickly extirpating non-heritable selfing. Because of that, mixed mating systems only lasted the complete set of 15,000 generations under the additive model, however, with very low stability.

#### 2.2.2. Heritable Selfing vs. Non-Heritable Selfing (See Methods)

The persistence of the two types of selfing was diametrically opposite. Alleles that conferred heritable selfing behaved as selfish genes and quickly became fixed in hundreds of generations, regardless of any kind of costs such as inbreeding depression and the *resistance cost of selfing*. An increase in the costs of inbreeding depression did not prevent the rise of the selfing genes and, if the costs were too high, they rather drove the population to extinction. In contrast, non-heritable selfing tended to become lost through costs brought about by inbreeding depression and the *resistance cost of selfing*. However, in the additive model, weak inbreeding depression enabled the persistence of selfing. 

#### 2.2.3. Cost and Benefits of Resistance and Tolerance

The costs of resistance and tolerance followed several theoretical predictions in the simulated systems. For instance, tolerance became fixed when *b_t_* > 0.5 > *c_t_* and lost if *b_t_* < 0.5 < *ct*. In addition, resistance tended to reach an allele frequency of 0.25 when costly, matching the predicted optimal. We summarized the main behavior of the simulated systems in a 2-D map (Figure 3). In the parametric space, the mixed defense strategies occurred in most of the areas where tolerance was not fixed because of the double trade-off between resistance and tolerance. Such a region was larger under the multiplicative model than under the additive model, also as predicted.

Regarding mixed mating systems, they were only present when resistance was present due to the *resistance cost of selfing,* which, in turn, only occurred when selfing was non-heritable. The mixed mating system was also more prevalent with the additive model, in which case it was present over the entire parametric space (Appendix A).

#### 2.2.4. Inbreeding Depression

The most striking observation regarding the mating system was that inbreeding depression actually failed to modulate selfing. Instead, selfing was affected the most by the type of model (multiplicative or additive) and the persistence of resistance and tolerance. The observed persistence of harmful alleles and average individual heterozygosis were congruent with the theoretical predictions. This is notable considering that they were subject to many sources of variability that were not included in the theoretical analysis.

The persistence of harmful alleles ranged from 0.0 to 5.0 (the predicted equilibrium value was 0.49) when selfing was fixed, and from 10.0 to 15.0 (the predicted equilibrium value was 9.74) when selfing was lost. The average individual genomic heterozygosis tended toward zero when selfing was fixed and toward 0.5 when selfing was lost (Appendix A).

The simulations also showed that dominance was the main cause of inbreeding depression in the short term, while overdominance was so in the long term, due to the efficient purging of harmful alleles in selfers.

#### 2.2.5. Linearity

The effects that linearity produced in the simulations resembled those produced by the type of model and the type of selfing. As equation analysis predicted, *l* > 1.0 enhanced the persistence of resistance and tolerance, and therefore also of mixed strategies. The converse occurred for *l* < 1.0 (Appendix A).

Linearity affected the mating system in two ways: Indirectly, via an effect on resistance and tolerance, which, in turn, affected selfing; and by buffering or amplifying the dominance effect of inbreeding depression. Linearity was so strong under the multiplicative model with non-heritable selfing that it led to the extinction or fixation of selfing (Appendix A).

#### 2.2.6. Special Cases

We investigated, by means of additional simulations and *in silico* experiments, certain simulations that presented unexpected and intriguing outcomes. The five most intriguing cases are summarized in Appendix A. They revealed a common interaction between the mating system and resistance by means of the *resistance cost of selfing,* as well as a tight connection between resistance and tolerance. Such a connection was, at times, positive, and at other times, negative, and presented correlations of up to 1.0 or −1.0. Some cases also presented behavior that was counterintuitive or apparently contradictory. For instance, in two special cases and in an extensive set of simulations, mixed strategies and a mixed mating system were unstable but persistent or recurrent in the long term, making it difficult to classify them as stable or unstable.

## 3. Discussion

We studied the effect of the mating system on the evolutionary stability of the mixed defense strategies of plants, first via a theoretical assessment of the equations that we created and then using ABM simulations. We observed a broad agreement between the simulation outcomes and the theoretical predictions, but all groups of simulations also showed unexpected outcomes. Those outcomes, when studied in detail, resulted from emerging behavior and path dependency that acted upon the effects of gene drift, the Hardy–Weinberg law, frequency-dependent selection, and interactions between the genotypes of plants and herbivores. Because of the complex interaction between these factors, the studied systems did not present infallible rules to predict the presence of mixed defense strategies and mixed mating systems. However, we found that some factors changed the fate of the simulated populations in radical ways while others changed it in a more subtle and gradual way. Because of that, and to better characterize the evolutionary stability of mixed defense strategies and mixed mating systems, we divided the relevant factors into qualitative factors and quantitative factors.

Qualitative factors had a large effect on populations, frequently producing opposite outcomes when changed. For example, heritable selfing consistently became fixed, while non-heritable selfing tended to become extinct. The type of model (additive and multiplicative) and the type of selfing (heritable and non-heritable) were qualitative factors.

Quantitative factors, in contrast, produced effects that accumulated gradually and affected the fate of resistance, tolerance, and the mating system in more complex ways than qualitative factors. The costs of tolerance and resistance, the benefit of resistance, and all parameters involved in inbreeding depression were quantitative factors.

Linearity behaved as a qualitative factor, flipping the fate of the simulated populations whether the value of the linearity parameter, *l*, was larger or smaller than one. However, as the value of *l* changed without crossing the 1.0 threshold, the populations showed incremental changes, making linearity also a quantitative factor.

### 3.1. Mixed Strategies

Our results suggest that plants’ resistance and tolerance to herbivory can interact with plants’ mating systems under a variety of conditions. However, the window of opportunity for mixed defense strategies was dependent on factors that interacted in sophisticated ways. In consequence, no factors were exclusively associated with the mixed strategies, although some improved the chances of mixed defense strategies. They could serve as a guide for predicting mixed defense strategies in natural populations. Those factors were:Multiplicativity. From the equation analysis, a multiplicative model predicts a concave surface (rounded downwards) of the fitness as a function of the resistance and tolerance costs. In natural populations, multiplicative accumulation of costs could occur if resource allocation to resistance or tolerance is proportional to the available resources of the plant and occurs sequentially (instead of simultaneously). This could occur, for instance, if the investment in resistance has a toll on growth and the investment in tolerance is “executed” later on when the plant has already a reduced size. Although multiplicativity can amplify the costs of mixed strategies, especially for plants with a mixed strategy, it can also enable the presence of mixed strategies as the proportionality of costs would always translate into non-zero fitness, in contrast to additivity, where the entire fitness could be cancelled.Non-linearity. A concave non-linearity of costs (by causes other than multiplicativity) or a concavity of benefits means that benefits would accumulate faster than costs, providing higher benefits to mixed strategies than to single strategies.High tolerance costs. Simulations showed that a cheap tolerance strongly inhibits resistance due to the mentioned double trade-off between resistance and tolerance. Therefore, only when the tolerance costs were high, was the resistance, and thus mixed strategies, able to occur.Low frequencies of heritable selfing. The *resistance cost of selfing* proved to be a ubiquitous mechanism that created a negative correlation between resistance and selfing. High selfing not only inhibited resistance in the simulations but also promoted the fixation of tolerance. Thus, low selfing promoted mixed strategies.Fluctuating selection of resistance traits. Frequency-dependent selection dynamics between resistance and anti-resistance resulted in cycling patterns in the simulations. In real populations, fluctuations are possible due to a number of mechanisms including “arms races”. In fluctuating scenarios, mixed strategies have wide windows of opportunity, especially but not exclusively during transitional stages when resistance declines due to unnecessary costs or decreased effectiveness.Gene-for-gene interactions between resistance and anti-resistance. Because the gene-for-gene model established optimal resistance frequencies as being low if resistance is costly, the Hardy–Weinberg law implies that a large proportion of the population remains unprotected by resistance, thus constituting a natural niche for tolerance.

Some features of the scenarios outlined above have been documented in natural populations in which mixed strategies were observed as well. For instance, many studies have related selfing to the effectiveness of resistance and tolerance [3,13,15,19,20,29,30,31,32,33,41,42]. We are aware that in all empirical studies, the trade-off between resistance and tolerance was not necessarily due to the mechanisms described here. However, our mechanisms can be partially met in some of those systems, and they can also serve as a guide to help explain more sophisticated relationships in the natural systems. For example, the testing of the resistance cost of selfing could be more useful for disentangling mechanisms than only correlating resistance traits and selfing.

### 3.2. Mating System

The use of different types of selfing caused the most contrasting behavior in the simulations. Heritable selfing quickly and consistently became fixed, regardless of inbreeding depression costs, whereas non-heritable selfing was consistently lost. This occurred because, when selfing was heritable, selfing genes behaved as genuine “selfish genes,” transferring high inbreeding depression costs to the organisms, but themselves retaining the “automatic” selection of selfing. This disentanglement produced an extreme outcome in which the genes walked relentlessly towards fixation even while populations moved towards extinction due to inbreeding depression costs.

In contrast with such extreme situations, most natural populations have presented variable and sometimes stable rates of selfing [43,44]. This implies that in natural populations with mixed mating, selfing is not a non-heritable selfing nor fully inherited selfing by a small number of genes [45,46]. Instead, it should have polygenic, epistatic, and plastic components. Despite that, our simulations accomplished their purpose of providing simple ways to account for how inbreeding depression interacts with defense systems. In fact, they constitute a demonstration of why natural populations do not present non-heritable selfing or a small number of selfing genes.

Our results suggest that sophisticated mechanisms could be involved in the maintenance of mixed defense strategies rather than simple balances of costs and benefits of resistance and tolerance. The observed interactions can provide guidelines for experimental studies at several levels. On a superficial level, studies could match their observations regarding mixed strategies with our proposed promoters of mixed strategies. If the observations match the expectations, specific mechanisms such as the resistance cost of selfing could be tested. A deeper understanding of such mechanisms could be of invaluable utility for designing effective strategies for dealing with herbivory in agricultural ecosystems.

## 4. Methods

First, we created a model integrating various features of previous models with new elements. Then, we performed an analysis of the equations to predict the parameter values that would promote mixed defense strategies and used those analyses altogether with pilot simulations to guide the construction of the experimental design of the main simulation runs. Finally, we carried out the ABM simulations to study the effect of the mating system on the evolutionary stability of mixed defense strategies. In ABM simulations, simulated agents represented organisms rather than populations, which enabled a realistic and statistically righteous assessment of various stochastic effects such as sampling error, gene drift, reproduction, and herbivore attacks on plants. Both the theoretical and simulation analyses were split into three parts to assess resistance and tolerance, inbreeding depression, and non-linearity. We also independently assessed additive and multiplicative forms of the model and two types of selfing.

### 4.1. The Model

We parametrized the tolerance and resistance costs following Simms and Rausher [38] and adopted the use of additive and multiplicative versions of the model as used by Restif and Koella [11] for the accumulation of the costs and benefits of resistance and tolerance (and inbreeding depression). In both models, the individual plant fitness (*w_i_*) is a function of the population base fitness (*w_0_*) adjusted by the cost of resistance (*RC*), the cost and benefits of tolerance (*TC* and *TB*, respectively), the damage by herbivory (*HD*), and the inbreeding depression provided by both overdominant (*O*) and dominant (*D*) effects:

Multiplicative model:(1)$$w_{i}=w_{0}\left(1-RC\right)\left(1-TC\right)\left(1-HD+TB\right)\left(O_{1}\right)\left(1-D\right)I_{\{uu=0\}}$$

Additive model:(2)$$w_{i}=w_{0}\left(1-RC-TC-HD+TB-O_{2}-D\right)I_{\{uu=0\}}$$
where the cost of resistance is $RC=c_{r}r^{l}$, the cost of tolerance is $TC=c_{t}t^{l}$, the cost of herbivory is $HD=c_{h}d_{\left(r,a\right)}^{l}$, the benefit of tolerance is $TB=b_{t}d_{\left(r,a\right)}^{l}t^{l}$, the overdominant component of inbreeding depression is $O_{1}={\left(\frac{h_{i}}{H}\right)}^{k}$, or $O_{2}=k\left(h_{i}-H\right)$, and the dominant component of inbreeding depression is $D={\left(u_{i}s\right)}^{l}$. The overdominant component is the ratio of individual heterozygosis to the overall heterozygosis of the population, which introduces an automatic adjustment given by heterozygosis: Individuals with above-average heterozygosis would have this term above one alongside a fitness increase, while below-average heterozygosis would shrink the fitness. The exponent becomes the scaling parameter that modulates this effect. In the additive model, this effect is given by simple subtraction. As for the dominant effect, the cost is obtained as the number of detrimental alleles multiplied by the selection coefficient, which is subtracted from 1.0. In the simulations, the number of detrimental alleles and amount of heterozygosis are obtained in a “genome” of 100 genes with a common mutation rate and selection coefficient.

The term $I_{\{uu=0\}}$ is the indicator function, which accounts for the lethality of harmful alleles when homozygous (i.e., =0). Variables are detailed in Table 3.

The non-linearly exponent *l* turns Equations (1) and (2) increasingly concave or convex as *l* departs from 1.0 (see Table 2). Notice that the terms involved in resistance and tolerance have a variable non-linearity, as do herbivory damage and inbreeding depression. As in previous models, we modelled tolerance as inducible [5,12,39], that is, the tolerance benefit is a function of the herbivore’s damage, $d_{\left(r,a\right)}$, which, in turn, depends on the interaction between resistance and anti-resistance genotypes in plants and herbivores.

The mating system is considered in Equations (1) and (2) through inbreeding depression (recall that Equations (1) and (2) refer to individual fitness). Since inbreeding depression results from the accumulation of deleterious alleles (dominant hypothesis) and, to a lesser degree, the loss of heterozygotes’ advantage (overdominance hypothesis) [21], we incorporated an overdominant component, given by the ratio of individual-to-base-population heterozygosis, and a dominant component, given by the accumulation of harmful alleles that were lethal when homozygous.

### 4.2. Equation Analysis and Predictions

To assess the opportunities of mixed defense strategies solely based on Equations (1) and (2), we derived the maximum points of Equations (1) and (2) when analyzed as functions of costs, benefits, and investments of resistance and tolerance. This analysis neglected the resistance benefits because they result from too many potential interactions of plant and herbivore genotypes (one of the motivations for using simulations). Despite that, it was possible to define scenarios that would promote mixed defense strategies from the theoretical point of view.

For the analysis of the effect of the mating system on mixed defense strategies, we calculated equilibrium points in which inbreeding depression was stable. Those points were functions of the number of harmful alleles (dominance) or the amount of individual heterozygosity (overdominance). Calculating equilibrium points allowed us to predict selfing costs for different regimes of mating systems.

The assessment of the effect of the non-linearity was more straightforward as it required us to analyze the relationship between the convexity or concavity of Equations (1) and (2) with the chances of mixed defense strategies.

### 4.3. Simulations

Each simulation created populations as collections of arrays (vectors) containing the relevant information of each organism. Each simulation contained two interacting populations, one of plants and one of herbivores, which evolved without plasticity, ontogeny, evolvability, or spatial heterogeneity. Populations were panmictic and diploid and had discrete generations. Plants were hermaphroditic, but herbivores were dioecious with synchronized generations. Herbivores attacked one plant per generation and succeeded in the attack by carrying the right genotype according to the gene-for-gene model, which is common in plants [40]. In this model, (1) plants without resistance alleles are fully susceptible to herbivory, (2) plants acquire full protection against herbivory by having at least one resistance allele, and (3) anti-resistance alleles in herbivores inactivate plant resistance, and one allele is enough to inactivate a plant’s resistance (for a given gene). For simplification, plant genotypes conferred complete or zero resistance while herbivory costs were constant. Plant fitness was adjusted by Equations (1) and (2) with the base fitness being randomly sampled from a Poisson distribution with λ = the base population fitness.

To avoid unnecessary complexity, we used two unlinked genes for controlling each effect among resistance, tolerance, selfing (in plants), and anti-resistance (in herbivores). Each gene was diallelic with one allele conferring the effect (e.g., resistance) and the other being neutral. The resources that plants invested in selfing, resistance, and tolerance (which were needed to scale the costs) were proportional to the amount of effect alleles being carried by each individual (also for herbivores). However, the tolerance in plants was also dependent on, and triggered by, herbivory.

We implemented two types of selfing: Heritable selfing, with all plants being self-compatible (and individual selfing ratios being proportional to the number of selfing alleles that were carried by the individual); and non-heritable selfing, where plants reproduced 100% by outcrossing or 100% by selfing with none having a mixed reproduction but both types being present in the population. Notice that selfing itself did not carry any cost, and inbreeding depression was generated as a byproduct of the accumulation of deleterious alleles or heterozygosis loss. To simulate this, each simulated plant contained, in addition to their effect genes (resistance, tolerance, and selfing), a “genome” of 100 genes with a common mutation rate and selection coefficient for the mutant deleterious alleles. Upon this genome, the heterozygosis of the offspring was calculated, and the segregated and newly mutated deleterious alleles were obtained. This modelling of the mating system allowed us to investigate the effects of inbreeding depression in a simple way.

The simulations were run in three groups, with each one assessing (a) the costs and benefits of resistance and tolerance, (b) inbreeding depression effects, and (c) non-linearity effects. In each group, we performed four sets of analyses for the combinations of the two models (additive and multiplicative) and the two types of selfing (heritable and non-heritable). Specifics of the simulation algorithm are given in Appendix A.

More than 2000 simulations were run for the main analyses, with 15,000 generations each. Additional simulations were run to establish the useful ranges of the parameters, assess the sensitivity of the simulations to initial conditions, test specific hypotheses, and investigate specific mechanisms behind certain results of interest.

## 5. Conclusions

The analysis of agent-based model simulations provided us with a unique opportunity to use the right amount of complexity to evaluate the interactions between the defense systems against herbivores and the mating system in plant populations. Specifically, the fitness of the simulated plants and herbivores resulted from their plant–herbivore interactions and the parameters governing tolerance, resistance, and inbreeding depression. In our computationally inefficient but realistic design, it was possible to observe interesting patterns of evolution such as the cryptic persistence and later resurgence of genes of resistance and selfing; boom-and-bust patterns of resistance, tolerance, or selfing; or long-term equilibria of those factors. Moreover, our agent-based simulations also showed pathological behaviors such as populations that collapsed due to inbreeding depression rather than purging their selfing genes, thus behaving as true selfish genes.

All in all, several factors correlated with mixed defense strategies and mixed mating systems. Some factors, such as concave linearity, multiplicative allocation of costs, and high tolerance costs, have relatively simple mechanisms and can be tested empirically. Other mechanisms, such as the double trade-off of resistance and tolerance and the resistance cost of selfing, provide specific mechanisms that can be proposed to explain empirical observations, e.g., a highly negative correlation between resistance and tolerance. Furthermore, the resistance cost of selfing may have large explanatory power for the widely observed negative correlation between resistance and selfing (or inbreeding). This could be true even for highly polygenic resistance apparatuses and ones that do not work under the gene-for-gene model because having low numbers of resistance alleles could be optimal to reduce the redundancy of polygenic resistance systems. Under those conditions, selfing would bring about extra costs of resistance. As a result of that, there is ample opportunity that the *resistance cost of selfing* exists in real populations of plants.

## Figures and Tables

**Figure 1 plants-12-00555-f001:**
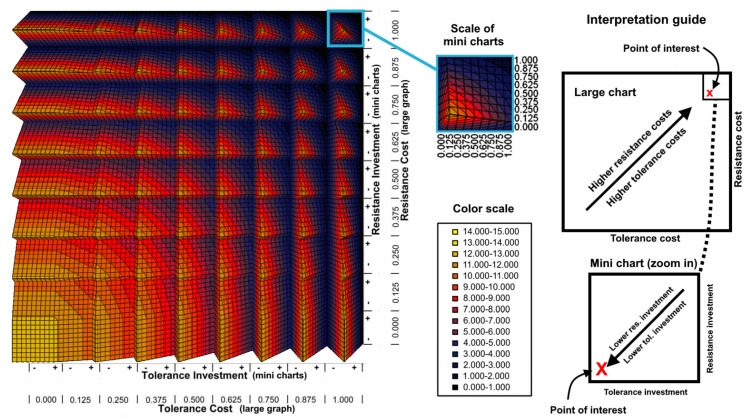
Fitness surface as function of resistance and tolerance. The charts show the fitness surface as a function of both costs and investments of resistance and tolerance. The large chart can be seen as the union of many mini charts that show a fitness surface as a function of resistance investment (vertical axis) and tolerance investment (horizontal axis) for a given combination of resistance cost (vertical axis, large scale) and tolerance cost (vertical axis, large scale). The tolerance benefit was set to 1-*c_t_* (e.g., 0.3 of tolerance cost also corresponds to 0.7 of tolerance benefit). This panel only represents the multiplicative model with non-herbivory (see the remaining combinations of multiplicative/additive and herbivory/non-herbivory in Appendix A). Fitness was defined as the overall number of seeds a population could produce in the absence of herbivores, and the costs and benefits are defined as proportions of fitness. Because resistance benefits depended on complex interactions with herbivores, they were not incorporated here. To facilitate interpretation, the drawing on the right shows the general tendencies in the large and small charts, with the “point of interest” having high resistance and tolerance costs (=1.000 for both) but low tolerance and resistance investments (=0.000, both), which result in a fitness value of 11.000–12.000 in the actual graph.

**Figure 2 plants-12-00555-f002:**
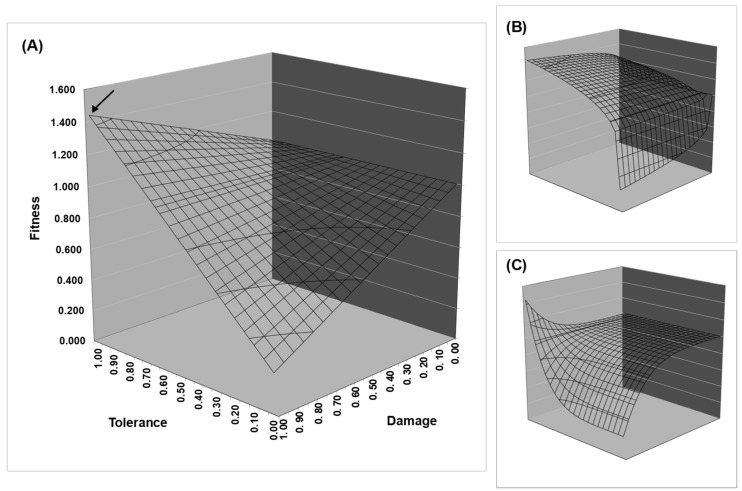
Surface of fitness as function of herbivory and linearity. Charts represent the herbivory term of Equations (1) or (2) as a fitness function of the damage caused by herbivory and tolerance investment. The displayed surfaces were obtained using different linearity values: (**A**) *l* = 1.0, (**B**) *l* = 5.0, and (**C**) *l* = 0.2. Observations: (*i*) Damage has a dual effect on herbivory by scaling the tolerance benefit and scaling the herbivore cost, so the surface could take several shapes for different tolerance values; (*ii*) shape depends on the values of *l* and also on the sign (negative or positive) of the variable, which is powered by *l* (see Table 2); (*iii*) moreover, tolerance produces different shapes for different damage values because, in the absence of damage, there is no tolerance benefit; (*iv*) the difference between the parameters *b_t_* (tolerance benefit) and *c_h_* (herbivory cost) can be seen as the point indicated by the arrow, which is equal to 1-*c_h_* + *b_t_*.

**Figure 3 plants-12-00555-f003:**
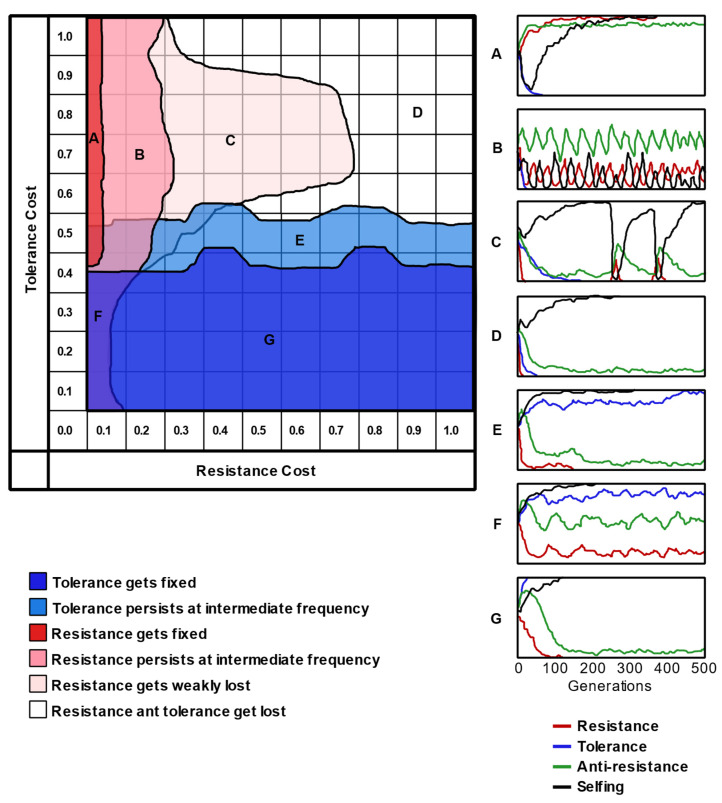
Zones of the parametric space of the resistance and tolerance costs/benefits. As previously, the tolerance benefit was equal to 1-*c_t_*. This division resulted from the analyses that are also displayed in Appendix A. Most of the zones were common to all four sets of simulations (two types of selfing × two models), and they differed only in the extent of the area that they occupied. On the right, examples of simulations corresponding to each zone are presented, showing the evolution of the frequencies (vertical scale) of resistance, tolerance, herbivores’ anti-resistance, and selfing over time (horizontal scale).

**Table 1 plants-12-00555-t001:** Theoretical predictions obtained from Equations (1) and (2) ^1^.

	Features	Multiplicative Model	Additive Model
Resistance Scenario	Max fitness	$w_{M}^{RS}\approx \left(1-c_{r}r^{*}\right)$	$w_{A}^{RS}\approx \left(1-c_{r}r^{*}\right)$
	Resistance	Only one allele retained	Only one allele retained
	Tolerance	Lost	Lost
	Key mechanism	Resistance persistence cancels herbivory damage and tolerance benefits	Resistance persistence cancels herbivory damage and tolerance benefits
	Relevant parameters	*c_r_*	*c_r_*
Non-Resistance Scenario	Max fitness	$\begin{array}{c} w_{M}^{NRS}\approx \left[1-t^{h}(b_{t}-c_{t})-c_{h}\right]\\ t^{h}=1_{\left\{0,1\right\}}\frac{1}{2}\left(\frac{1}{c_{t}}-\frac{1-c_{h}}{b_{t}}\right) \end{array}$	$w_{A}^{NRS}\approx \left[1-t^{*}(b_{t}-c_{t})-c_{h}\right]$
	Resistance	Lost	Lost
	Tolerance	Persists at variable frequencies	Persists at variable frequencies
	Key mechanism	Lack of resistance enables damage and thus tolerance benefits	Lack of resistance enables damage and thus tolerance benefits
	Relevant parameters	*c_t_ c_h_ b_t_*	*c_t_ c_h_ b_t_*
Fitness ratio (*fr*) between scenarios		$f_{r}=\frac{w_{M}^{NRS}}{w_{M}^{RS}}$	$f_{r}=\frac{w_{A}^{NRS}}{w_{A}^{RS}}$

^1^ Equations were analyzed as functions of the costs and investments of resistance and tolerance, as well as the benefits of tolerance. The symbol *r^*^* refers to the minimum possible investment in resistance without disabling resistance, which occurs when only a single resistance allele is retained. The symbol *t^*^* is the maximum investment in tolerance, and 1_{1,0}_ is the indicator function, which takes the value of one if the term at left belongs the interval (0.0–1.0), and zero otherwise.

**Table 2 plants-12-00555-t002:** Effects of the linearity parameter *l*.

*k·a^l^*	*k·a* > 0.0	*k·a* < 0.0
*l* > 1.0	Convex (˅)	Concave (˄)
*l* < 1.0	Concave (˄)	Convex (˅)

**Table 3 plants-12-00555-t003:** Symbols in Equations (1) and (2).

Symbol	Description	Notes	Range	Default Values ^a^
*w_i_*	Individual fitness	Number of seeds	0–25	N/A
*w_0_*	Base population fitness	Number of seeds	25	25.0
*c_r_*	Maximum cost of resistance	It is a proportion	0.0–1.0	0.15
*r*	Investment in resistance ^b^	It is a proportion	0.0–1.0	N/A
*c_t_*	Maximum cost of tolerance	It is a proportion	0.0–1.0	0.4
*t*	Investment in tolerance ^b^	It is a proportion	0.0–1.0	N/A
*c_h_*	Maximum cost by herbivory	It is a proportion	0.0–1.0	0.75
*d* _(*r,a*)_	Herbivory damage	It is a proportion	0.0–1.0	N/A
*b_t_*	Maximum benefit of tolerance	It is a proportion	0.0–1.0	0.25
*k*	Over-dominance parameter	Promotes the effect	0.0–1.0	0.3
*h_i_*	Individual genomic heterozygosis	Over 100 genes	0.0–0.5	N/A
*H*	Overall heterozygosis of the population	It is a constant	0.0–1.0	0.25
*u_i_*	Individual number of detrimental alleles	Accumulated	0.0–∞	N/A
*s*	Selection coefficient (for detrimental alleles)	Loss of fitness	0.0–1.0	0.005
*l*	Linearity parameter	Shape parameter	0.2–5.0	1.0
*I* _(*uu*=0)_	Indicator function for the presence of Homozygous lethal alleles	=0 when homozygous =1 otherwise	-	N/A
*c_a_*	Maximum cost of the anti-resistance (herbivores)	It is a proportion	0.0–1.0	0.4
*a*	Investment in anti-resistance (herbivores)	It is a proportion	0.0–1.0	N/A

^a^ The default values correspond to those used in the simulations. ^b^ Investments in resistance and tolerance are determined by the proportion of alleles that confer the effect in the genotype.

## Data Availability

The source code and a graphic-interface app can be found in: https://1drv.ms/u/s!AgDuZ9Kt2jykx11sOLIgw6ToBcaN?e=0AjBbc (accessed on 19 January 2023); https://1drv.ms/u/s!AgDuZ9Kt2jykx1zlQgEISiawjm0e?e=5igFe4 (accessed on 19 January 2023).

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
