# Peer review of "The Joint Evolution of Herbivory Defense and Mating System in Plants: A Simulation Approach"

_plants, 2023, doi:10.3390/plants12030555_

Round 1

Reviewer 1 Report

The authors report a mathematical analysis that simulates mating systems on the evolutionary stability of plants.

In my opinion, the manuscript is not suitable for PLANTS. In fact, I have not found ideas applicable to botanical sciences. Perhaps, the manuscript could have potential for mathematical journals.

The most critical aspect concerns the concept of mating in plants. The authors should know that plant species have multiple mating and interaction strategies with the environment such as mosses, ferns, conifers, or the vast group of Angiosperms. How can the authors propose a predictive model if the variety of reproductive strategies is so varied? This important criticality is accompanied by the absence of model validation. The authors have not validated their model which may not find application in any plant species or only in the most complex plant species.

Author Response

Dear Reviewer,

We have included in a single, ordered, document the response to all reviewers.

Sincerely,

the authors.

Reviewer 2 Report

Dear Authors 

The present manuscript entitled "The joint evolution of herbivory defense and mating system in plants: A simulation approach" discuss how non-linearity and mating system affect the evolutionary stability of mixed defense strategies by  means of agent-based model simulations. The Abstract and introduction are well written with focused informations. Methods are described in details and results are quite impressive, and discussed nicely. 

There is a need to include a short conclusion at the end which may explain the key findings of the study in a simple way. The conclusion may be helpful for the readers from a different area.

With regards

Author Response

(The authors gave the same response as above.)

Reviewer 3 Report

In this manuscript, the author established a new method based on the previous model to study and calculate the effect of the mating system on the evolutionary stability of the mixed defense strategies of plants. Although there are many articles reporting on the energy consumption of plant defense response, few research so far have specifically addressed the factors influencing the costs and benefits of plant defense.  But there are also some problems with the paper. 

1.     The article says there are two main strategies that plants adopt against herbivores infestation. I suggest the authors should explain the strategies of plants to defend themselves against insects, such as physical means: coating, waxy layer; the chemical way: hormonal pathways, etc.

2.     The format of the references like "[e.g., 14-18]","[see reviews 3, 14, 15]"(line 45/59 in the manuscript) should be modified following the journal’s style.

3.     The author should apply the algorithm to verify the biological function to show the reliability of the algorithm.

4.     Whether the author considers the effects of plant insect interaction?

5.     When using the new method, the authors should distinguish whether there are some differences in the effects of plants on different insects. For example, the differences in feeding by chewing insects and sucking insects may affect the degree of plant defense?

Author Response

(The authors gave the same response as above.)

Reviewer 4 Report

The simulation study by Sandoval-Castellanos & Nunez-Farfan investigates the effect of the mating system on the effect of herbivory, taking into account both tolerance and resistance to herbivory, and both using cost  and benefits  parameters. The authors use two models, an additive model and a multiplicative model, which produce similar scenarios, although the former produces a convex surface of fitness, whereas the latter produces a flat surface.Their model represents an extension of previous related models on the costs and benefits of plant response to disease and herbivory, yet providing some further interesting insights.

My major concern is the use of totally heritable selfing and non-heritable selfing which produce drastic scenarios, e.g. rapid fixation of selfing alleles, population extinction, in one case, or rapid loss of non-heritable selfing in the second case. As the author pointed out, in natural populations, selfing is neither totally non-heritable, nor specified by a small number of genes; their scenarios are extreme, and not representative of the biological world. A range of outcrossing rates should be used, and complete selfing avoided, since even highly selfing species occasionally outcross (see for example Platt et al. PLoS Genetics 6: e1000843, 2010).

Furthermore, looking at the equations for the additive and multiplicative model, what are the parameters specifying the two types of selfing? The authors state that the mating system is expressed as inbreeding depression (the term O1  and O2); however,  it is not clear how the two types of selfing translate into the amount of inbreeding depression. Please provide a clearer explanation. 

I also don’t particularly like figure 1: values on the axis are too small, and the figures are difficult to read and understand. I think a simplified version should be used, and a more detailed one given in the supplementary file.

Author Response

(The authors gave the same response as above.)

Round 2

Reviewer 1 Report

The authors discussed and argued well the critical issues that I had exposed in the previous review. I now recognize the importance of their simulation work on plant defense strategies.